# Granulomatous Reactions Following the Injection of Multiple Aesthetic Microimplants: A Complication Associated with Excessive Filler Exposure in a Predisposed Patient

**DOI:** 10.3390/reports8040194

**Published:** 2025-09-30

**Authors:** Marjorie Garcerant Tafur, Carmen Rodríguez-Cerdeira

**Affiliations:** 1Bexclinic, Guzmán el Bueno, 53, 28015 Madrid, Spain; 2Department of Health Sciences, University CEU San Pablo, Julián Romea 23, 28003 Madrid, Spain; 3Fundación Vithas, Grupo Hospitalario Vithas, Príncipe de Vergara 109, 28002 Madrid, Spain; 4Dermatology Department, Grupo Hospitalario (CMQ Concheiro), Manuel Olivié 11, 36203 Vigo, Spain; 5Department of Health Sciences, Campus of Vigo, University of Vigo, As Lagoas, 36310 Vigo, Spain; 6European Women’s Dermatologic and Venereologic Society (EWDVS), 36700 Tui, Spain

**Keywords:** granulomatous reaction, dermal fillers, autoimmune predisposition, GPA, calcium hydroxyapatite, polynucleotides, non-invasive treatment, ultrasound diagnosis

## Abstract

**Background and Clinical Significance**: Granulomatous reactions are rare but clinically significant complications of aesthetic procedures involving dermal fillers, particularly in individuals with underlying immune dysregulation. These reactions present diagnostic and therapeutic challenges, especially when associated with undiagnosed or latent autoimmune diseases. This case illustrates the interaction between filler composition, immune status, and the risk of delayed inflammatory responses, underscoring the need for thorough patient evaluation and individualized management strategies. **Case Presentation**: A 49-year-old woman developed delayed-onset subcutaneous nodules following midface augmentation with two filler types: a monophasic, cross-linked hyaluronic acid gel (concentration 20 mg/mL, 1.0 mL per side) injected into the deep malar fat pads, and a calcium hydroxyapatite suspension (30% CaHA microspheres in a carboxymethylcellulose carrier, 0.5 mL per side) placed in the subdermal plane along the zygomatic arch. The procedure was performed in a single session using a 22 G blunt cannula, with no immediate adverse events. High-resolution ultrasound demonstrated hypoechoic inflammatory nodules without systemic symptoms. A retrospective review of her medical history revealed a latent, previously undisclosed diagnosis of granulomatosis with polyangiitis (GPA). The immune-adjuvant properties of calcium hydroxyapatite likely triggered a localized pro-inflammatory response in this predisposed patient. A conservative, staged, non-invasive therapeutic protocol—saline infiltration, intradermal polynucleotide injections, and manual lymphatic drainage—achieved complete clinical and radiological resolution without systemic immunosuppression or surgical intervention. **Conclusions:** This case highlights the critical importance of pre-procedural immunological assessment in aesthetic medicine. Subclinical autoimmune conditions may predispose patients to delayed granulomatous reactions after filler injections. An individualized, conservative management strategy can effectively resolve such complications while minimizing the risks associated with aggressive treatment. Greater awareness of immune-mediated responses to dermal fillers is essential to ensure patient safety and optimize clinical outcomes.

## 1. Introduction and Clinical Significance

In recent decades, minimally invasive aesthetic procedures have shown exponential growth worldwide, largely due to their perceived safety, low morbidity, and accessibility. Among these, dermal filler materials have become a key tool in volume defect correction, facial rejuvenation, and the treatment of asymmetries or post-traumatic sequelae, as well as in therapeutic indications such as oncologic reconstruction; congenital malformations; and other conditions affecting the musculoskeletal, ophthalmologic, or urogenital systems [1].

Dermal fillers are classified according to their durability as permanent, semipermanent, or biodegradable. They can also be divided into reversible substances (such as hyaluronic acid) or collagen stimulators (such as poly-L-lactic acid and calcium hydroxyapatite) [2,3,4].

Although their use has been widely promoted as safe and associated with only transient side effects, the scientific literature has reported an increasing number of both local and systemic complications, some of which may present in a delayed fashion, appearing weeks, months, or even years after the initial injection [5].

These delayed reactions encompass a broad spectrum, ranging from localized inflammation to more complex immune-mediated processes [6].

Histologically, granulomatous reactions show chronic infiltrates with macrophages and giant cells, and despite claims of biocompatibility, fillers may act as immune adjuvants triggering local or systemic responses [7].

The causes of these reactions are multifactorial and include the chemical nature of the injected material; the administration technique; the cumulative antigenic load; and, fundamentally, the immunological characteristics of the host [8].

Among the most relevant risk factors are latent or subclinical autoimmune conditions, which may have gone undiagnosed or may be in clinical remission while maintaining an underlying state of immune hyperreactivity. In such individuals, fillers can trigger pathological reactivations by disrupting the previously established immunological balance [9].

Filler materials may persist as foreign bodies, inducing a chronic innate immune response through pro-inflammatory mediators [10], sometimes manifesting as slowly evolving granulomas mimicking neoplasms, infections, or autoimmune processes [11].

Delayed immune reactions may be precipitated by infections, vaccinations, surgery, or systemic stress [12]. Granulomatosis with polyangiitis, a systemic necrotizing vasculitis formerly called Wegener’s granulomatosis, exemplifies how an autoimmune disease can interfere with fillers [13]. It primarily affects the respiratory tract and kidneys; is associated with anti-neutrophil cytoplasmic antibodies, particularly c-ANCA against proteinase 3 [14]; and its pathophysiology involves a type III hypersensitivity reaction with immune complex formation, complement activation, and endothelial injury [15].

Untreated, the mortality rate of this condition is extremely high [16]. Management requires aggressive immunosuppressive therapy, including glucocorticoids and cyclophosphamide for remission induction, followed by maintenance agents, such as methotrexate or azathioprine. It is essential to recognize that even during clinical remission phases, these patients may exhibit an exaggerated immune response to external stimuli, such as biomaterials, potentially triggering a localized or systemic disease reactivation [17].

Histologically, filler-induced granulomas typically show mononuclear infiltrates, foreign body-type giant cells, fibrosis, and sometimes exogenous material within vacuoles [18,19].

Recent studies have employed non-invasive imaging techniques, such as high-resolution cutaneous ultrasound and elastography, to identify features consistent with these reactions, including poorly defined hypoechoic nodules or echogenic structures with posterior acoustic shadowing [20].

The differential diagnosis of these lesions includes mycobacterial infections, cutaneous sarcoidosis, panniculitis, and cutaneous lymphomas [21].

In this context, a thorough medical history is crucial, encompassing the autoimmune background, previous immunosuppressive treatments, the vaccination history, and exposure to filler products [22].

Unfortunately, the lack of standardized protocols and limited awareness among healthcare professionals regarding the potential interaction between dermal fillers and immunologic disorders have contributed to an underestimation of the associated risks, leading to a higher incidence of adverse reactions in vulnerable patients [23].

Management of granulomatous reactions remains difficult due to the lack of standardized guidelines, with approaches ranging from conservative local therapies (saline injections, corticosteroids, polynucleotides, lymphatic drainage) to systemic immunosuppression or surgical excision in refractory cases [24,25,26]. These complications highlight the risks of increasing medicalization in aesthetics and stress the importance of holistic management, including immunological assessment, interdisciplinary collaboration, and continuously updating pathophysiological mechanisms [27,28,29,30].

In this context, the present work aims to review the pathophysiology of granulomatous reactions associated with dermal fillers, their potential link to latent autoimmune conditions, and the current management strategies. Through a critical analysis of the literature, we seek to provide a scientific foundation for safer and more responsible aesthetic practice—one that integrates immunological knowledge with dermocosmetic techniques.

## 2. Case Presentation

We present the case of a 49-year-old woman from Colombia with Fitzpatrick skin phototype IV–V who was referred to our private clinic for evaluation of a localized facial complication following an aesthetic procedure.

The investigation was conducted in accordance with the principles outlined in the Declaration of Helsinki (1975, revised in 2013). According to point 23 of this declaration, approval must be obtained in compliance with the regulations governing private aesthetic medicine practices in Spain, the documentation of which is attached. The study was approved by the Bexclinic management and signed by the patient on 25 June 2025.

Ethical review and approval of this study were not required by the Institutional Review Board/Ethics Committee of Bexclinic (C/Guzmán El Bueno 53, 28015, Madrid) because case reports are not considered research. The patient’s information has been de-identified.

Several weeks earlier, the patient had undergone a facial treatment with dermal fillers, including one hyaluronic acid vial (sodium HA 0.9%, very high molecular weight 2700–3400 kDa) and one 3 mL vial of a collagen stimulator with the following formulation: 30% synthetic, biocompatible, and resorbable calcium hydroxyapatite (CaHA) microspheres, 25–45 microns in diameter, and a 70% aqueous carrier gel composed of sterile water, glycerin, and carboxymethylcellulose (CMC) as a thickening agent for facial rejuvenation and volume restoration in the midface region.

The procedure was initially uneventful; however, approximately two weeks later, the patient noted a progressive, firm, painless swelling in the right malar region, which did not resolve spontaneously.

At presentation, the patient was afebrile and exhibited no systemic symptoms. Physical examination revealed a subcutaneous nodule approximately 3 cm in diameter in the right malar area, accompanied by two smaller nodules in the submalar region (Figure 1). All lesions were firm, mobile, and non-tender and showed no signs of erythema, local warmth, or fluctuation. The patient’s overall condition was good.

High-resolution cutaneous ultrasound demonstrated homogeneous hypoechoic images compatible with well-demarcated subcutaneous inflammatory infiltrates, with no evidence of fluid collections or necrosis.

Ultrasound imaging revealed alternating columnar hypoechoic areas and distributed linear hyperechoic zones. There were disruptions in the normal echoanatomy, as evidenced by diffuse hypoechogenicity starting at the basal dermis, which obscured the distinction of deeper anatomical planes and was associated with a slight increase in microvascularization.

In the right lateral submalar area, poor visualization of the soft tissue echoanatomy was noted, with uniform isoechogenicity extending from the posterior dermis to the bone interface. This finding is likely related to infiltration by dermal filler material that was poorly integrated into the surrounding tissue (Figure 2 and Figure 3).

Notably, there is a markedly thickened subepidermal low-echogenicity band (SLEB), which is likely indicative of obstructive lymphedema.

During an extended anamnesis, the patient disclosed a personal history of chronic granulomatous vasculitis, specifically granulomatosis with polyangiitis (GPA). This condition, formerly known as Wegener’s granulomatosis, is a necrotizing vasculitis that affects small- and medium-sized vessels and is characterized by granuloma formation and variable systemic involvement. The patient had a confirmed diagnosis for several years, with intermittent flare-ups and ongoing rheumatologic follow-up. At the time of the filler injection, she was not under active immunosuppressive therapy but had failed to disclose this information during the pre-procedural aesthetic consultation.

This finding was pivotal in establishing the diagnosis of a delayed granulomatous inflammatory reaction secondary to filler materials in a patient with an underlying immunological predisposition. Considering this, a conservative, stepwise therapeutic protocol was implemented.

First stage: Forty cubic centimeters of isotonic saline solution were injected on several occasions at a depth of 1.0–1.5 cm using a fanning technique. The objective was to mechanically disperse the residual filler material and reduce the local antigenic load. Injections were administered bilaterally in the malar region, targeting the subcutaneous plane with a 22G blunt-tip cannula.

Second stage (after 15 days): The procedure was repeated using the same volume, this time at a consistent depth of 1.5 cm. This approach resulted in a significant reduction in the nodular volume and a marked improvement in the facial contour.

Third stage: Intradermal administration of polynucleotides was performed to promote tissue regeneration and mitigate residual inflammation. The protocol comprised 4 mL of highly purified sodium DNA solution (two 2 mL ampules) combined with 2.5 mL of highly purified sodium DNA gel (one 2.5 mL vial), for a total of 6.5 mL per session. This regimen was applied every two weeks for three sessions.

Additionally, specialized facial lymphatic drainage was prescribed to improve the tissue circulation and facilitate the resorption of infiltrated material.

The clinical course was notably favorable (Figure 4). Six weeks after initiation of treatment, both the primary and secondary nodules had resolved clinically and sonographically. There was no need for systemic corticosteroids, antibiotics, immunosuppressants, or surgical intervention. The patient achieved complete recovery of facial symmetry, with no visible aesthetic sequelae or recurrence during follow-up (Figure 5). She was discharged with specific recommendations to avoid further injectable treatments.

## 3. Discussion

Previous reports have described adverse reactions to facial fillers, including sarcoid-like granulomas in patients under interferon therapy and other injection-related complications [31,32].

In this case, the injection of a collagen biostimulator in the patient with granulomatosis with polyangiitis in remission may have triggered a localized granulomatous response [33,34,35,36]. According to Kim et al., cross-linking agents, such as 1,4-butanediol diglycidyl ether, can increase the immunogenicity of hyaluronic acid [37]. Narins et al. reported on granulomas with poly-L-lactic acid, whose delayed collagen-stimulating effect may elicit disproportionate immune reactions in predisposed individuals [38]. The combined use of these fillers may amplify local immune activation, while calcium hydroxyapatite, described by Breithaupt et al., has strong biostimulatory properties and induces type I collagen [39,40].

However, like other semipermanent fillers, it may trigger delayed granulomatous reactions, particularly in patients with an immunologically altered background. From a histopathological perspective, calcium hydroxyapatite (CaHA)-induced granulomas exhibit a foreign-body-type reaction characterized by epithelioid macrophages, multinucleated giant cells, and perilesional fibrosis [41].

During inflammatory phases, these reactions have been documented to appear on ultrasound as well-defined hypoechoic areas, sometimes with posterior acoustic shadowing, which is consistent with the findings observed in our patient [42].

The role of CaHA as an immunological adjuvant may have amplified the local inflammatory response, acting synergistically with PLLA on a previously sensitized immune system [43].

Clinically, granulomatous reactions exhibit a variable spectrum. In some cases, they present as firm, painless nodules without overt inflammatory signs, while in others, they are accompanied by erythema, induration, and occasionally ulceration [44].

According to Feller-Heppt et al. [25], clinical diagnosis can be challenging and is often mistaken for infections, panniculitis, or foreign body reactions.

Cutaneous ultrasound has emerged as a valuable tool in this context [25,45,46,47].

As noted by Scotto et al. [48], filler-induced granulomatous nodules typically present as homogeneous hypoechoic patterns, often with posterior shadowing or internal vacuoles that reflect the shape of the infiltrated particles.

While high-resolution ultrasound was highly valuable in supporting the clinical suspicion and guiding the therapeutic decisions, histopathological confirmation would certainly have strengthened the diagnosis. Nevertheless, considering the facial location of the lesions and the potential risk of scarring or aesthetic compromise, a conservative diagnostic approach was deemed more appropriate than performing a biopsy.

Therapeutic management of these reactions remains a matter of ongoing debate. No standardized protocols exist, and decisions are generally based on the clinician’s experience and patient’s tolerance. The preferred treatment often includes systemic or intralesional corticosteroids, 5-fluorouracil [49], hyaluronidase (in cases involving HA), empirical antibiotics, ultrasound therapy [49], immunomodulators such as methotrexate, a pulsed laser [50], or even surgical intervention [51].

However, these approaches are not without risk. In the present case, a conservative, stepwise, non-pharmacological strategy was adopted, involving physiological saline infiltration, intradermal polynucleotides, and manual lymphatic drainage. This approach proved effective, allowing complete resolution without the use of immunosuppressants or surgical procedures.

Regarding the use of polynucleotides, their application in aesthetic medicine has attracted growing interest. These compounds exhibit anti-inflammatory, biostimulatory, and regenerative properties, promoting tissue repair without triggering systemic adverse effects. In this context, their use allowed for modulation of persistent inflammation and facilitated dermal matrix remodeling [52].

Although less documented, infiltration with physiological saline has been used as a mechanical strategy to disperse persistent fillers, where Kanchwala et al. [53] have reported its usefulness in resistant chronic granulomas. Evidence for manual lymphatic drainage remains largely anecdotal, yet it appears to be a safe adjunct for reducing edema and enhancing tissue circulation [54]. This case highlights key clinical recommendations: the need for thorough medical evaluation prior to aesthetic procedures, especially in patients with immunological histories—even in remission—as minor stimuli, such as vaccinations or cosmetic treatments, may trigger adverse reactions [55]. Moreover, informed consent should explicitly address autoimmune diseases, prior immunomodulatory therapies, and rheumatologic history since patients often fail to consider these relevant to cosmetic care [56]. Overall, this clinical experience aligns with the international literature, emphasizing the importance of assessing the patient’s immunological status before dermal filler use and favoring individualized therapeutic strategies that prioritize tissue physiology and local immunomodulation.

## 4. Conclusions

This case highlights the importance of a careful medical evaluation prior to aesthetic procedures with immunomodulatory materials and illustrates how an incomplete anamnesis may lead to significant clinical consequences, even in patients who otherwise appear healthy. The favorable outcome observed suggests that an individualized and conservative therapeutic approach can be effective in selected situations while minimizing systemic risk. Although based on a single case, these findings support the potential benefit of an interdisciplinary collaboration between dermatologists, rheumatologists, and aesthetic practitioners in addressing complex complications and enhancing patient safety.

## Figures and Tables

**Figure 1 reports-08-00194-f001:**
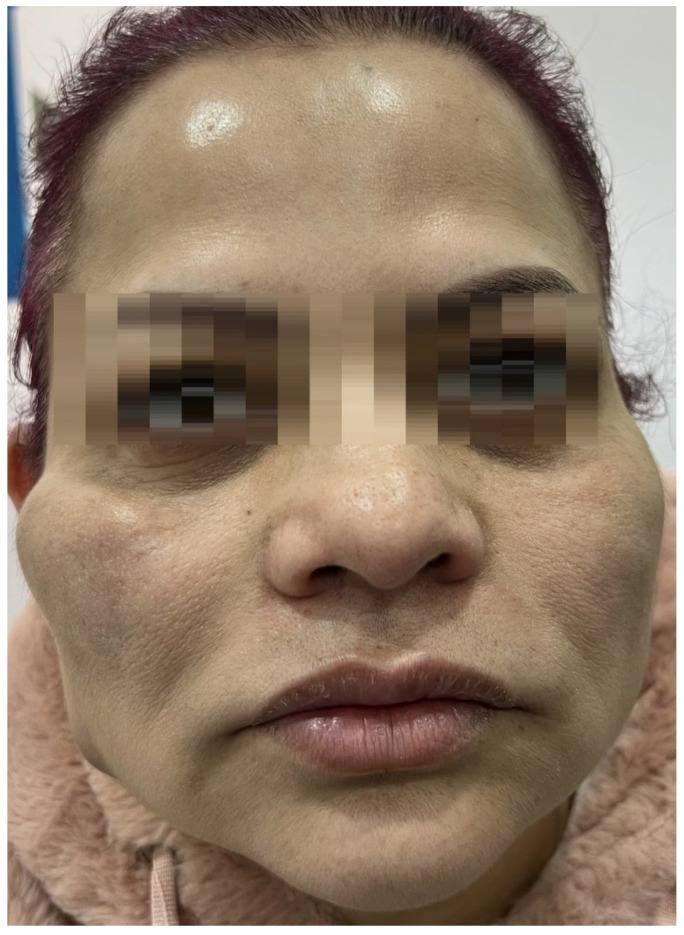
At the time of presentation, the patient exhibited firm, palpable, non-tender, and disfiguring nodules.

**Figure 2 reports-08-00194-f002:**
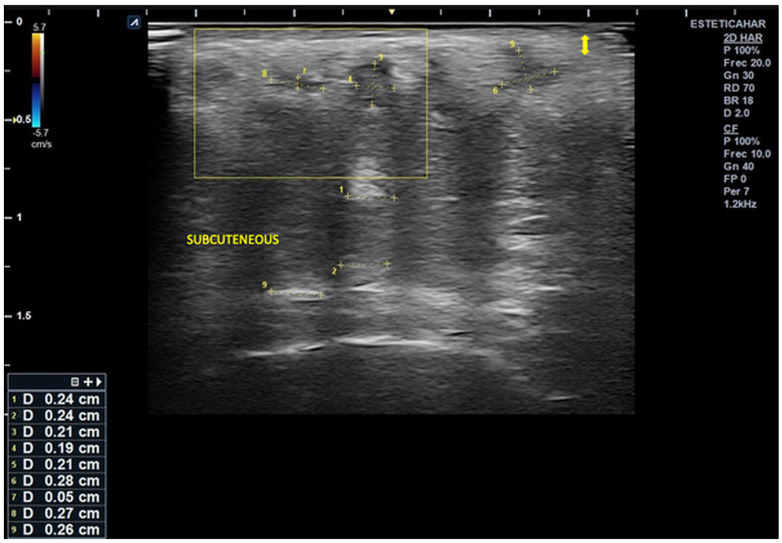
Ultrasound image of the right lateral malar region demonstrating destructured epidermis and dermis (

). The scan reveals alternating hyper- and hypoechoic columnar areas extending from the posterior dermis to the bone plane, which hinder clear differentiation of tissue layers based on echogenicity. The largest area, measuring 0.24 cm in width, is most likely associated with deposits of hybrid material (hyperdiluted collagen stimulator). Additionally, three smaller, rounded hypoechoic deposits are observed in the more superficial subcutaneous region. The largest of these measures 0.27 × 0.26 cm and is likely associated with poor integration of the injected material, without evidence of increased microvascularization.

**Figure 3 reports-08-00194-f003:**
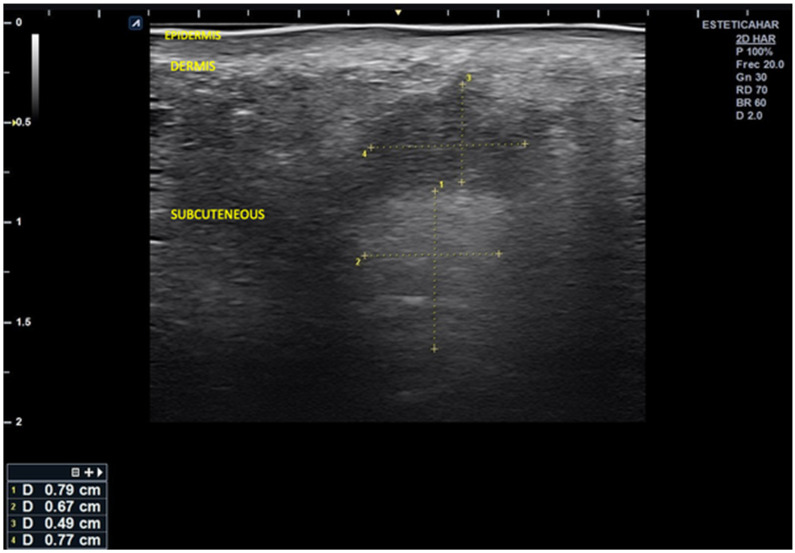
Image of the center-right malar region showing disruption of the normal layered echoanatomy, with two well-defined, rounded areas. A superficial subcutaneous hypoechoic area measuring 0.49 × 0.77 cm is likely associated with the liquid component of the injected material. An underlying hyperechoic area measuring 0.79 × 0.67 cm is probably related to the solid portion of the hybrid filler.

**Figure 4 reports-08-00194-f004:**
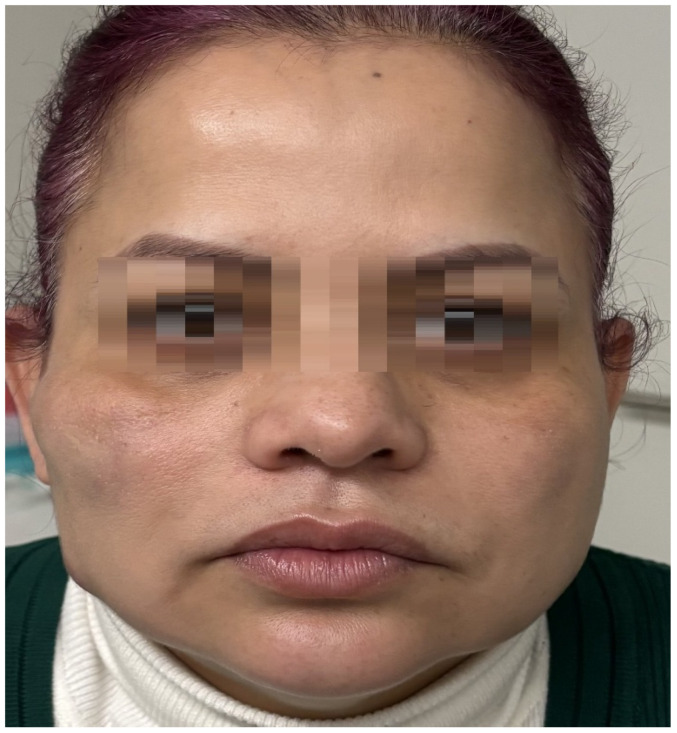
Eight days after the first treatment session, the patient showed initial clinical improvement.

**Figure 5 reports-08-00194-f005:**
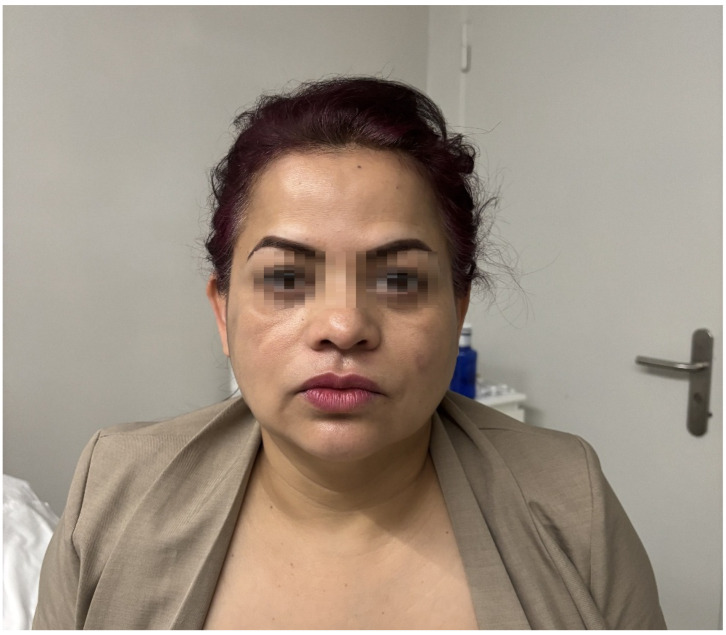
Two months after the treatment, the patient’s condition demonstrated significant resolution of the lesions and restoration of the facial contour.

## Data Availability

The original contributions presented in this study are included in the article. Further inquiries can be directed to the corresponding author.

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
