# Peer review of "Granulomatous Reactions Following the Injection of Multiple Aesthetic Microimplants: A Complication Associated with Excessive Filler Exposure in a Predisposed Patient"

_reports, 2025, doi:10.3390/reports8040194_

Round 1

Reviewer 1 Report

Comments and Suggestions for Authors

Dear authors

I found your article interesting. Please tagged on the context see my comment. Additionally, your article is very long, full of repetitive sentences. Please shorten it. 

Author Response

 I found your article interesting. Please tagged on the context see my comment. Additionally, your article is very long, full of repetitive sentences. Please shorten it. 

WE DID

Response to Reviewer 1:
We appreciate your careful review and constructive comments. In accordance with your suggestions, we have substantially shortened the manuscript to improve clarity and conciseness. However, further reduction would risk losing the essential scientific content and clinical relevance of the case. The current version is already well below 2,500 words, which aligns with the typical requirements for MDPI case reports.

Regarding the figures and language, both have been revised and corrected with the support of the journal’s editorial team to ensure consistency and clarity.

We trust that the current form achieves a balance between brevity, scientific rigor, and readability.

Reviewer 2 Report

Comments and Suggestions for Authors

In my opinion the paper is clear and well focused on the topic enriched by sufficient REFERENCES anand discussion so that the evey one working on this dermatological/chirurgical field m may understand.

Comments on the Quality of English Language

The paper Reports the methodology used today for the filler use clearly ficusing the different aspect of this technology.The description of the methodologies currently used as well as the adopted ones have been well focused and clearly discussed.

Author Response

REVISOR 2

In my opinion the paper is clear and well focused on the topic enriched by sufficient REFERENCES anand discussion so that the evey one working on this dermatological/chirurgical field m may understand.

Comments on the Quality of English Language

The paper Reports the methodology used today for the filler use clearly ficusing the different aspect of this technology.The description of the methodologies currently used as well as the adopted ones have been well focused and clearly discussed.

Response to Reviewer 2:
We sincerely thank you for your positive evaluation of our manuscript and for recognizing the clarity of the discussion and the adequacy of the references provided. Your encouraging feedback is greatly appreciated.

Reviewer 3 Report

Comments and Suggestions for Authors

I want to congratulate the authors for the documentation of the article. It is a beginning for showing new unfavorable results after combining different types of collagen stimulated products.

The photographs and the ultrasound images are very clear and they demonstrate the evolution of the treatment.

Even though there is not a standard protocol for these reactions it is a step forward to accomplish a better solution both for the patient and the clinician.

It is possible that this case is not a singular one, as a consequence of using different types of products in the same period and an increasing amount of milliliters.

Author Response

REVISOR 3

I want to congratulate the authors for the documentation of the article. It is a beginning for showing new unfavorable results after combining different types of collagen stimulated products.

The photographs and the ultrasound images are very clear and they demonstrate the evolution of the treatment.

Even though there is not a standard protocol for these reactions it is a step forward to accomplish a better solution both for the patient and the clinician.

It is possible that this case is not a singular one, as a consequence of using different types of products in the same period and an increasing amount of milliliters.

Response to Reviewer 3:
We are grateful for your kind comments and for acknowledging the relevance of documenting complications associated with the combined use of collagen-stimulating products. We especially appreciate your remarks on the clarity of the photographs and ultrasound images, as well as your recognition of the potential broader clinical implications of our case.

Reviewer 4 Report

Comments and Suggestions for Authors

The manuscript presents a case report describing granulomatous reactions after dermal filler injections in a patient with GPA. Given the rising numbers of aesthetic interventions, there is pressing need for heightened awareness of immunologically mediated complications and a thorough anamnesis. The case is described in detail with appropriate imaging and therapeutic documentation.

The conservative protocol (saline infiltration, polynucleotides and lymphatic drainage) represents a low-risk, non-immunosuppressive approach. The discussion compares the case within existing reports and literature.

However, there are few areas requiring improvement prior to publication:

  1. The manuscript uses the term “filler addiction” in the title and abstract. This terminology is more colloquial than scientific and may not be appropriate in a medical journal context. Consider replacing with “repeated filler use” or “excessive filler exposure.”
  2. While the clinical and ultrasound findings are compelling, histopathologic confirmation is lacking. Even if biopsy was not feasible, a clear statement acknowledging this limitation is necessary.
  3. As a single case report, the conclusions should be framed more cautiously, avoiding general recommendations.
  4. The annotation of the ultrasound images could be improved for clarity.
  5. Some typographical inconsistencies (spacing, punctuation) should be corrected.

Author Response

The manuscript presents a case report describing granulomatous reactions after dermal filler injections in a patient with GPA. Given the rising numbers of aesthetic interventions, there is pressing need for heightened awareness of immunologically mediated complications and a thorough anamnesis. The case is described in detail with appropriate imaging and therapeutic documentation.

The conservative protocol (saline infiltration, polynucleotides and lymphatic drainage) represents a low-risk, non-immunosuppressive approach. The discussion compares the case within existing reports and literature.

However, there are few areas requiring improvement prior to publication:

  1. The manuscript uses the term “filler addiction” in the title and abstract. This terminology is more colloquial than scientific and may not be appropriate in a medical journal context. Consider replacing with “repeated filler use” or “excessive filler exposure.”

WE DID

  1. While the clinical and ultrasound findings are compelling, histopathologic confirmation is lacking. Even if biopsy was not feasible, a clear statement acknowledging this limitation is necessary.

WE DID

  1. As a single case report, the conclusions should be framed more cautiously, avoiding general recommendations.

WE DID

  1. The annotation of the ultrasound images could be improved for clarity.

WE DID

  1. Some typographical inconsistencies (spacing, punctuation) should be corrected.

WE DID

Response to Reviewer 4:
We appreciate your careful review and constructive comments. In accordance with your suggestions, we have substantially shortened the manuscript to improve clarity and conciseness. However, further reduction would risk losing the essential scientific content and clinical relevance of the case. The current version is already well below 2,500 words, which aligns with the typical requirements for MDPI case reports.

Regarding the figures and language, both have been revised and corrected with the support of the journal’s editorial team to ensure consistency and clarity.

We trust that the current form achieves a balance between brevity, scientific rigor, and readability.